# Social determinants of health among noncitizen deported US veterans: A participatory action study

**Frances Tao[1], Cassidy T. Lee[2], Edgar Castelan[3], Ann Marie Cheney[4]***

**1** Department of Family Medicine, University of California Los Angeles, Los Angeles, California, United States of America, **2** School of Osteopathic Medicine in Arizona, A.T. Still University, Mesa, Arizona, United States of America, **3** California State Senate, Office of State Senator Connie M. Leyva, San Bernardino, California, United States of America, **4** Department of Social Medicine Population and Public Health, School of Medicine, University of California Riverside, Riverside, California, United States of America

* ann.cheney@medsch.ucr.edu

**Data Availability Statement:** The datasets used and/or analyzed during the current study are available via the Qualitative Data Repository: Cheney, Ann. 2023. "Data for: Health of Deported

## Abstract

This qualitative study examines the social determinants of health among noncitizen deported United States veterans. We utilized Photovoice, a participatory action research method used to inform structural level change, with 12 veterans. Audio-recorded semi-structured interviews explored photos and discussed deportation's effects on veteran health. We performed rapid template and matrix analysis of interview transcripts. Interviews were conducted in Tijuana, Mexico from December 2018 to January 2019. Study findings show that veterans prioritize returning to the United States to improve their quality of life. Analysis of photos and narrative text indicated that deportation caused social, economic, and political insecurities. Veterans struggled to maintain access to necessities post-deportation. Disrupted social networks compounded their situation, resulting in chronic stress and poor health outcomes. The findings from this study offer insight into the ways deportation acts as a social determinant of health. The findings suggest modifying veteran reintegration programs, as well as reforming criminal justice and immigration laws, such as creating more Veteran Treatment Courts and allowing immigration judges to consider military history during deportation proceedings involving noncitizen veterans.

## Introduction

A common misnomer in American society is that service in the United States military guarantees a fast track to citizenship for noncitizen servicemembers. While military service provides a pathway to citizenship, it does not guarantee automatic citizenship nor does it guard against deportation [1–3]. Since World War I, US Congress has provided a pathway for noncitizen servicemembers serving honorably in the US military during wartime [4] and most recently in peacetime [5]. In theory noncitizen servicemembers can apply for citizenship immediately following basic training in times of war and after one year of service (continuous or cumulative); yet, in practice this does not play out. Less than 50% of eligible servicemembers naturalize during military service [1]. Too often servicemembers mistakenly leave the military believing they

Non-citizen US Military Veterans." https://doi.org/10.5064/F6OLPRYD.

**Funding:** Research reported in this publication was supported by the University of California Riverside School of Medicine Dean's Innovation Award, the UC MEXUS Graduate Student Award, The Dean's Innovation Award and Graduate Student Award provided funding for data collection, including participant incentives, materials and supplies, and travel costs.

**Competing interests:** The authors have declared that no competing interests exist.

are naturalized US citizens only to realize post discharge they are not citizens and therefore not afforded the protections that accompany citizenship [6].

The transition from the military to the civilian world is abrupt and veterans often struggle to assimilate to civilian life post-discharge [2, 7–10]. Compared to civilian citizens, veterans experience poorer physical and mental health, which may contribute to increased difficulty maintaining employment, homelessness, strained interpersonal relationships, substance use, and legal trouble [9, 10]. Many cope with lingering effects of combat-related violence and trauma, placing them at risk for unhealthy and even unlawful behaviors as well as incarceration [1–3].

This transition is especially precarious for noncitizen veterans. Immigration and criminal law changes in the 1980s and 90s expanded deportable offenses to include nonviolent misdemeanors (e.g., shoplifting and drug possession) [1, 3]. Criminal charges, once considered minor, now serve as the basis for exile. Additionally, immigration judges no longer consider length of US residency and military service as reasons against deportation [3]. These policy changes result in the punishment of noncitizen veterans who risk banishment from the very country they risked their lives to protect.

Since the mid-1990s, the US has deported thousands of noncitizen veterans, including veterans who committed misdemeanors such as disorderedly conduct or drunk driving [11]. Noncitizen deported US military veterans comprise a vulnerable and often unrecognized health disparity population. Latinx of Mexican and Central American origin are overrepresented in this deported veteran population [2, 3, 12], and over 80% of deported veterans report medical issues and approximately 75% lack access to healthcare after deportation [3].

In this paper, we examine the effects of deportation on the health and wellbeing of noncitizen veterans who served in the US military and identify veteran-informed solutions to address deported veterans' healthcare needs. We utilize the socio-ecological model (SEM) to guide our research questions and analysis. This framework describes how national level policies, infrastructure (e.g., healthcare systems), and interpersonal factors (e.g., social networks) intersect with individual beliefs and practices to shape health outcomes [13]. Building on what others have done, we consider how deportation as a social determinant of health influenced by the multiple, intersecting layers of the SEM impinges upon individual-level health outcomes among noncitizen Latinx veterans deported to Mexico.

## Methods

This research was carried out from December 2018 to January 2019. The research stems from initial conversations with deported veterans accessing free medical services in Tijuana, Mexico through the mobile medical clinic Healing Hearts Across Borders. This qualitative study used photovoice, a participatory action method that uses photography and narrative text to voice the experiences of vulnerable populations and contribute informative dialogue towards structural level intervention [14]. We chose this method as it has been used to successfully encourage war veterans to reflect on past trauma and can bring the intimate experiences of veterans into public spaces to enact change [15].

Using a community-engaged research (CEnR) approach, we obtained veteran involvement and input to begin to address the healthcare and social needs of this vulnerable veteran population [16]. A multidisciplinary team of academics in medicine and public policy partnered with the Deported Veterans Support House (DVSH) to conduct the research. In preparation for the research and in line with CEnR approaches, we held a community review board, a one-time meeting used to engage deported veterans in the research process by eliciting their input into research questions, study design, and execution of the research plan [17]. We also saw the review board as a way to develop trust with deported veterans [18].

All research was approved prior to the start of research by the University of California Riverside Institutional Review Board. The IRB approval reference number is HS-18-094 dated 7/18/18. Approval by a relevant Mexican institutional ethics committee was not required; rather, the IRB required an access letter from the partnering organization in Mexico, the Deported Veteran Support House, that acknowledged their involvement in the research and outlined the study's goals, research activities, and involvement of veterans in their organization (see S1 Text). Written informed consent was obtained from all study participants involved in the photovoice research activity, including permission to use their photographs for analysis and dissemination purposes. Verbal consent was obtained from participants in the community review board; participants were provided with an information sheet outlining the project, risks and benefits of participation, and study and IRB contact information. The institutional IRB provided a waiver of written informed consent for this group interview as the consent document would have been the only thing to connect veterans to the study.

## Setting

Research was conducted at the DVSH in Tijuana, Mexico. Heralded as a "deportation city" given its proximity to the United States–Mexico border, Tijuana is home to the busiest land border crossing in the world and houses two deportation stations where individuals are handed over to Mexican officials [19–21]. The DVSH is a nonprofit organization whose mission is to support and advocate for deported veterans and their families [22]. It functions as a shelter and resource center. The DVSH has served over 100 noncitizen US military veterans deported to over 25 countries around the world.

## Sampling and recruitment

We worked closely with the DVSH to disseminate study flyers and spread study information via word of mouth for purposive and snowball participant recruitment. Eligible participants were at least 18 years of age, US veterans of any service era, Mexican citizens by birth, US residents for at least 10 years, deported to Mexico in the last 20 years, and currently residing in Tijuana, Mexico.

## Sociodemographic survey

During an initial in-person meeting, we administered a brief sociodemographic survey to obtain key sociodemographic characteristics of the sample, including age at interview and age upon entry to the US, prior US legal status, branch of the miliary (Army, Marine Corps, Coast Guard, Navy, Air Force), year enlisted and discharged, deployment, military discharge status (e.g., administrative, honorable, dishonorable), and injuries/disabilities sustained during military service. This survey also inquired about perceived difficulty assimilating back to civilian life after miliary discharge, including year deported, jail-time prior to deportation, Spanish proficiency pre-deportation, and legal proof of Mexican citizenship post-deportation. Last, participants were asked to prioritize with 1 being of highest priority the following items in respect to their quality of life: food, shelter, income/employment, healthcare, social support, returning to US/US citizenship. Responses were analyzed for descriptive statistics.

## PhotoVoice elicitation

At the initial in-person meeting, we provided participants with instruction on how to use photography to symbolize their experiences and needs as deported veterans. We provided each participant with a camera or the option to use their own camera and instructed them to take

photos of images that captured their responses to the following prompts: 1) "What does daily life look like for you?" 2) "How is your life impacted by deportation?" 3) "What are some things that impact your health?" 4) "What are some things you would like help with or want changed?" 5) "Who would you like for us to share this information with and what do you want them to know?", and 6) "What do you want US policy makers to know so that other veterans aren't in your situation?" Participants were provided 1-week to take photos and asked to select 10 photos that best represent their responses to the prompts. They were also asked to follow guidelines to protect their own identities and those of others. For instance, when taking pictures in public spaces, participants were instructed to anonymize people in those spaces by ensuring faces or physical identifiers (e.g., tattoos) were not visible. They were also instructed to avoid documenting risky situations to ensure their safety.

## Semi-structured interviews

Trained interviewers conducted in-person semi-structured interviews with participants using an open-ended interview guide with probes to elicit information on deported veterans' experiences and healthcare and social needs.

The interviewer first asked participants to share their thoughts on the following question: "What is it like to be a US veteran and a Mexican citizen having been deported from the US?" using probes to elicit additional information about participants' experiences and understandings of the relationships between deportation and health. Next, interviewers asked participants to consider the photos they took and select two or three photos that best represent their experiences as a deported veteran. As participants narrated the meaning of the photo, the interviewer used probes to understand how social determinants of health, including housing, employment, financial stability, transportation, and access to resources such as healthcare, impacted their health status. Then, interviewers asked participants to select another two to three photos that best represented ways to support deported veterans and to narrate the meaning of the image. Last, interviewers asked participants to comment on how the images can be used to inform policymakers about deported veterans' experiences and social and health needs. Interviews were conducted in English, audio-recorded, and professionally transcribed. Interviews lasted between 60 to 120 minutes.

## Data analysis

A total of 14 veterans participated in the research; however, only 12 of the 14 veterans completed the sociodemographic survey, photo elicitation, and semi-structured interview and were included in the analysis. Our sample, which included all male veterans deported to Tijuana, Mexico affiliated with the DVSH, shared common deportation and health experiences. Thus, we stopped data collection once data saturation was achieved. For nonprobability samples, data saturation for the main research questions, meaning no new information or themes emerge, can be reached within 12 interviews with a homogenous sample [23].

A rapid analytic approach that blended deductive and inductive analysis was used. Transcripts were first summarized using template analysis, which involves creating a template populated by key topics in the interview guide [24, 25]. Analysts identified key points, inserted responses into the template by topic, and summarized the transcripts. Matrix analysis was then used to identify themes across templates. This approach organizes the summaries of each template by participants (rows) and key topics (columns) to facilitate the identification of patterns and emergent themes across participants' experiences [24]. The SEM framework was used to interpret the meaning of the themes in relationship to deportation and health outcomes and illustrative quotes were used to evidence the patterns.

## Results

### Participant characteristics

As indicated in Table 1, 12 male veterans completed all phases of the study. The average age was 56 years at time of participation and 6 years at entry into the US. All reported previous lawful permanent resident status. Participants served in the US Marine Corps (50.0%), Army (33.3%), and Navy (16.7%). One third (33.3%) saw combat. Two-thirds (66.7%) sustained injuries and/or disabilities during service. Over half (58.3%) reported at least some difficulties assimilating back to civilian life post-discharge. All were incarcerated prior to deportation. Half of the participants reported that returning to the US was most important for their quality of life (Fig 1). Securing income (25%) and accessing healthcare (17%) were the next highest priorities. Social support was among the least important.

**Table 1. Participant sociodemographic characteristics.**

| Characteristic (n = 12) | | n (%) | mean, SD |
|---|---|---|---|
| Sex, Male | | 12 (100.0) | |
| Age | | | 56.42 (11.85) |
| Age when first entered USA | | | 6.58 (5.64) |
| Prior US legal status = Lawful Permanent Resident / Green Card Holder | | 12 (100.0) | |
| Branch of US military | | | |
| | Marine Corps | 6 (50.0) | |
| | Army | 4 (33.3) | |
| | Navy | 2 (16.7) | |
| Year enlisted | | | 1981.08 (10.18) |
| Year discharged | | | 1985.50 (11.13) |
| Years of service | | | 4.42 (2.11) |
| Military discharge status | | | |
| | Honorable | 9 (75.0) | |
| | Other than Honorable | 3 (25.0) | |
| Participants who had combat duty | | 4 (33.3) | |
| Participants who sustained injuries during military service | | 8 (66.7) | |
| Difficulty assimilating back to civilian life after military discharge | | | |
| | None of the time | 5 (41.7) | |
| | A little of the time | 2 (16.7) | |
| | Some of the time | 3 (25.0) | |
| | Most of the time | 1 (8.3) | |
| | All of the time | 1 (8.3) | |
| Year of deportation | | | 2008.25 (6.97) |
| Jailed prior to deportation | | 12 (100.0) | |
| Proficient in Spanish at the time of deportation | | | |
| | Strongly disagree | 1 (8.3) | |
| | Disagree | 3 (25.0) | |
| | Neither disagree nor agree | 2 (16.7) | |
| | Agree | 2 (16.7) | |
| | Strongly agree | 4 (33.3) | |
| Legal proof of Mexican citizenship | | 11 (91.7) | |

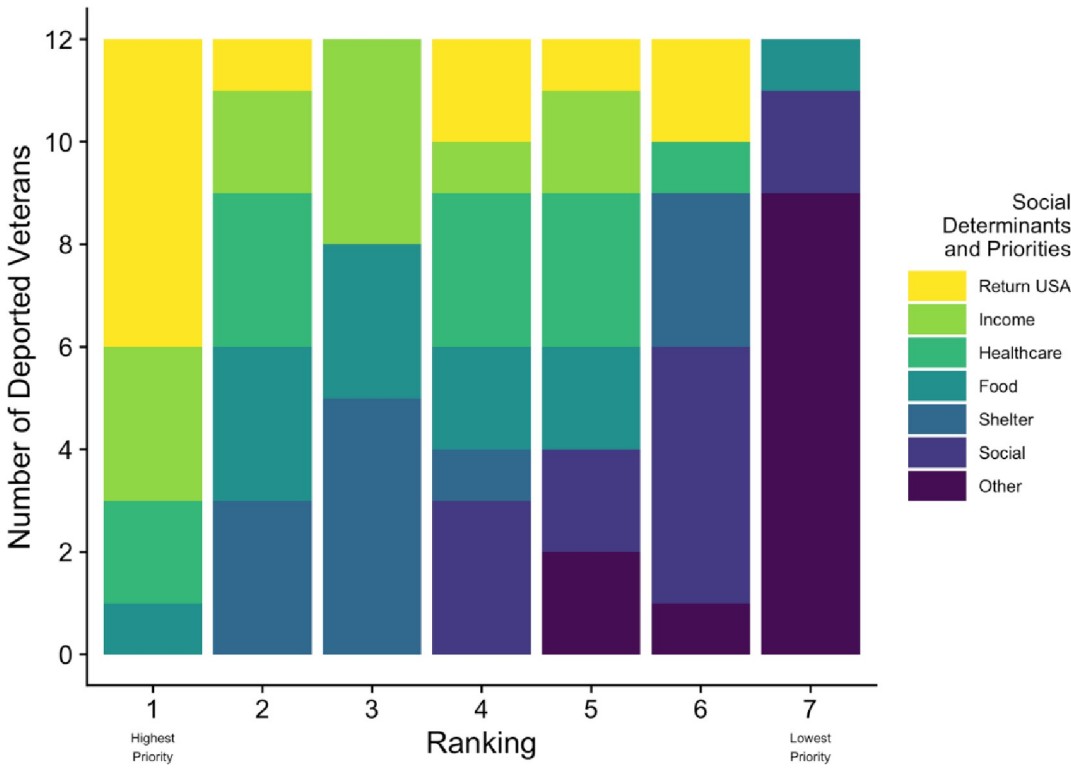

**Fig 1. Socio-Ecological Model (SEM).**

## Intersections between the SEM layers

As depicted in Fig 2, our findings highlight how deportation plays out in each sphere of influence of the SEM, placing some groups and individuals in structurally vulnerable positions which contributes to chronic stress and harms individual health [26, 27]. Broad political and

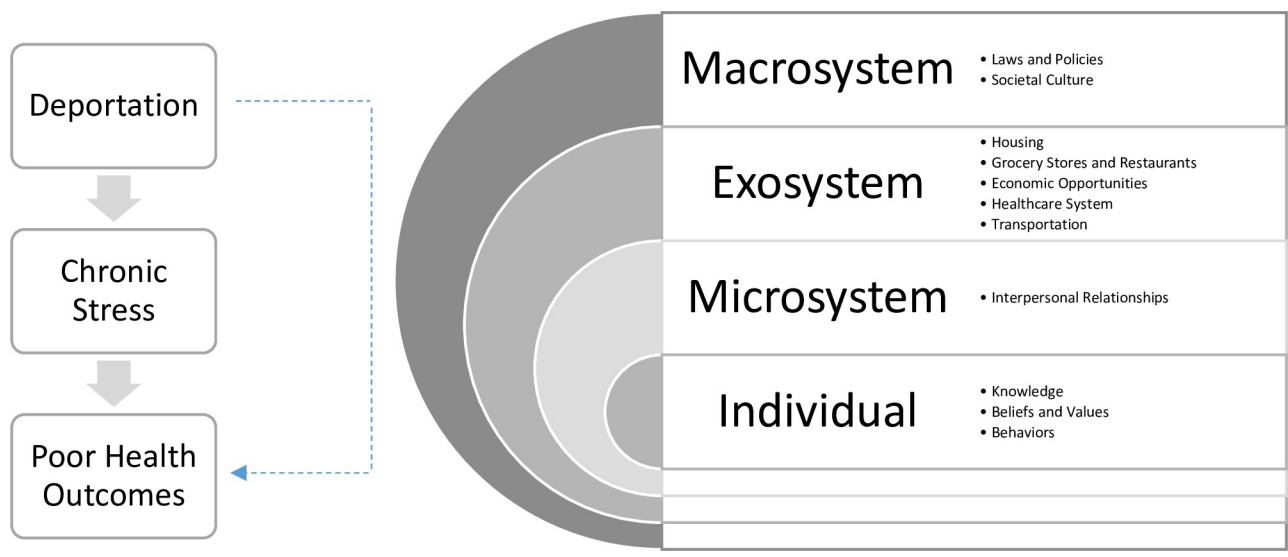

**Fig 2. Self-reported priorities of non-US citizen deported military veterans.**

cultural processes at the macrosystem, infrastructure at the exosystem (housing, food security, economic opportunities, transportation, and healthcare systems), and interpersonal and social networks at the microsystem shape deportation and intersect with individual-level factors creating chronic stress and resulting in poor health outcomes. Feelings of loneliness, despair, and physical pain were often managed through substance use.

## The macrosystem: Political and sociocultural processes

As veterans in our study discussed, deportation displaced veterans from their homes and families and forced them to live out their daily life in different communities and environments. Deported veterans' narratives shed light on how they uniquely stand in a position where political and sociocultural elements in both the country they were born (e.g., Mexico) and the country they were raised (i.e., US) reject them. Veterans voiced disbelief about their situation, viewing deportation as a betrayal or ongoing punishment by the US government. Yet, they remained loyal to the US. Many indicated they would serve again in the US military.

**Displacement.** Veterans in our study unanimously self-identified as "American" and believed the US is their home. Unable to reconcile their new reality, veterans described their situation as being stuck in "limbo." Their lack of Spanish proficiency hindered their ability to assimilate, compounding their alienation. "We're not educated, trained, or have the same beliefs as here [in Mexico], so we can't really adapt. . . We're just surviving here. Because we have no choice," reflected a Marine Corps veteran who lived his entire life in the US. Some believe they are rejected from their own families. A Vietnam-era Navy veteran recalled his extended family members "would give me this [judgmental] look, 'Oh, you got deported?'" Participants felt abandoned. They lost touch with loved ones in the US. "We're getting forgotten," a veteran said.

## The exosystem: Infrastructures

Displaced to a new country, veterans in our study found themselves confronted with novel infrastructures they described as "chaos." Deportation resulted in loss of food, housing, and medical care. Veterans understood to maintain access to basic needs, they required stable income. The pressure of day-to-day survival created chronic stress.

**Access to basic needs.** Post-deportation, veterans in our study were homeless and food insecure. They relied on the charity of others (e.g., homeless shelters, soup kitchens, and free health clinics) to survive. They juggled basic needs with limited income. Caring for his family of four, a Marine Corps veteran said, "You'll spend all your check just in medicine or in doctor consultations and you won't be able to pay your rent, groceries, or your bills." Narrating his experiences of deportation, a 75-year-old Army veteran shared via a photo of a grocery store— he explained having to exchange items at the pawn shop for money to buy food (S1 Data). These harsh realities evoked fear among veterans in our study. A physically healthy 39-year-old Army veteran remarked, "There's no future for me. . . I gotta do something quick or eventually it's gonna be me out there begging for money."

**Employment.** Veterans reported challenges finding and securing jobs in Mexico. They attributed their struggles to negative perceptions of deported veterans, limited Spanish proficiency, unreliable transportation, and age discrimination. Some beliefs influence hiring practices, such as the belief that noncitizen veterans betrayed Mexico by serving in the US military and are criminals due to deportation. Older veterans also experienced age discrimination. Being viewed as a liability to companies, some were passed over for job offers. Most veterans lived job to job. Some secured a steady income via call center employment where their fluency in English was valued (S2 Data).

**Transportation.**  Deported veterans cited public transportation issues as a common barrier to employment. Buses take indirect routes with incomplete coverage of cities. They also make infrequent and missed stops. The cost is prohibitive especially if veterans take multiple buses. Commenting on his photo, a 63-year-old Marine Corps veteran with chronic pain explained the difficulties of public transit: "Some people actually take three buses to get to work. . . It's a hit and miss every single time" (S3 Data). Similarly, a 61-year-old Navy veteran recalled the impossible math of poverty: "It would cost me about 100 pesos a day [to ride the bus to work], which would be like $10 [USD]. . . But it's $10 I couldn't afford. . . I would have to hitchhike to go to work." Those who relied on public transportation cited how it exacerbated physical health and chronic pain. They reported injuries sustained from poorly maintained buses and potholes they walked across to get to bus stops. Referring to a photo of a sidewalk, a Vietnam-era veteran said, "I've seen people fall, and I've twisted an ankle. . . I have screws in an ankle, so if I step on it wrong, oh, man, the pain is just really bad" (S4 Data).

## The microsystem: Interpersonal relationships

Deportation physically separated veterans from their loved ones, placing stress on their relationships. Navigating a novel society and infrastructure forced veterans to establish new networks. Some forged friendships with others deported from the US, but not all were successful in doing so.

**Strained interpersonal relationships.**  Veterans often could not contact their families for extensive periods of time. They struggled with guilt. Being deported was "depressing" and "traumatic." One Marine Corps veteran who lived in the US for over 50 years explained how he could handle unstable housing and lower income in Mexico, but the hardest part of his deportation was isolation. "Being away from my family. . . I go home and I'm alone" (S5 Data). Many veterans experienced loneliness and were unable to fulfill family roles. A 44-year-old Navy veteran recalled how it felt to be unable to see his son for 15 years: "[It is] torture in itself for me and for [my son]. . . for many, many years, he was upset with me. He was thinking that I abandoned him."

Veterans recreated social networks in Mexico, befriending others whom they met in homeless shelters or at work. They believed it was easiest to build friendships with those who were deported from the US because they shared the same trauma and endured similar struggles. "All of the colleagues that I work with are deported. . . [and so] I feel right at home," explained a 75-year-old Army veteran, deported for nearly a decade.

## Individual health

The stress of survival and disruption of social networks took a toll on veterans' health. Chronic illness and disease, such as hypertension and diabetes, were common and often unaddressed. Referring to a photo of his body (S6 Data), a Navy combat veteran with chronic internal bleeding said:

> I'm not making enough to buy whatever private [health insurance] thing they have. . . I just paid [the doctor] for the consultation and he told me, 'Hey, you need so much more.' But I can't afford to do any of that. So, little by little, I'm [physically] breaking down.

Veterans frequently experienced depression related to physical separation from families and loneliness, as well as anxiety due to feelings of helplessness and despair. A veteran reflected:

We've been left in limbo and unsure of our future. That's breeding depression, anxiety, [and] a whole load of mental issues. . . the fear that I have is that I'm gonna fall into a depression that I won't be able to come out of.

Faced with aberrant circumstances void of strong social support, veterans find themselves with thoughts of death and engage in risky behaviors.

Many veterans used substances to cope with deportation. A 49-year-old Army veteran, who lived in the US since infancy, started drinking alcohol when he was deported. "When I first got here [in Mexico]. . . I gave up on everything. . . and I really didn't care if I died anymore. It didn't matter." A Marine Corps veteran who sustained military-related injuries that affected his daily life shared how wine took "away all the problems for a moment" (S7 Data). "You have to forget here," he said. Others attempted to quit and succeeded but could easily relapse. A Marine Corps veteran in recovery shared, "I know I'm still susceptible, because it's a disease and I just got it under control. . . To just think that I could drink a six pack or something, I'd just be fooling myself."

## Discussion

Deportation is more than a single political event resulting in the transfer of an individual from one country to another. As the narratives of veterans in our study show, it is a stressful life event [19, 28] shaped by sociocultural, economic, and political processes that create stress and contribute to poor health outcomes. Using the SEM, we saw how political and sociocultural factors displaced noncitizen veterans. While they perceived themselves as Americans, the US government did not. Within their country of origin (i.e., Mexico) they felt as outsiders unable to speak the language and navigate the infrastructure of daily life from attending to their basic needs (housing, food) to finding employment to getting around (transportation). We also saw how deportation stressed their interpersonal relationships creating tension with family in the US and pushing many to establish new networks in Mexico. The intersection of the macrosystem (political and sociocultural processes), exosystem (housing, food, employment, transportation), and microsystem (interpersonal relationships, networks) with individual-level experiences of deportation contributed to chronic stress and poor physical and mental health outcomes and substance use as a coping mechanism. The findings provide insight into possible intervention.

First, upstream factors or changes at the macrosystem that would address political and sociocultural processes that underlie the deportation of US veterans need to be addressed. Unlike other deported populations, noncitizen veterans are intimately tied with the US government. Deported veterans voiced disbelief about how the US could dispose of them after they risked their lives to protect their country. Changes in policy is one way to address noncitizen veterans' deportation. For instance, politicians recently introduced the Veterans Pathway to Citizenship Act of 2021 [26], an amendment to the Immigration and Nationality Act that would extend the time in which noncitizen veterans could apply for citizenship. Strict enforcement of policy implementation is another way. The US Government Accountability Office report of Immigration and Customs Enforcement (ICE) of noncitizen veterans found that ICE did not consistently collect and maintain data on veteran status and thus did not follow policy that would have considered veterans' service history in decisions to remove veterans from the country [27].

Second, changes can be made with the exosystem including infrastructures both in the US and abroad. Additional Veteran Treatment Courts (VTCs) are needed to minimize veterans' entry into civilian criminal justice systems. VTCs provide a means for rehabilitative justice

that allows appropriate treatment of mental illness while reducing recidivism [1, 28, 29]. Similarly, healthcare systems can better treat this veteran population. Many veterans seek healthcare services outside of the Department of Veterans Affairs [30]; as such healthcare providers across healthcare systems can work together to reduce risk for deportation by developing competencies and curricula to foster the knowledge and skills to elicit military history, identify veteran-specific exposures, and provide evidence-based treatments for service-related health conditions (e.g., PTSD, substance use) that could induce risky behavior leading to incarceration. Given that a high correlation exists between military service, mental illness, substance abuse, and crime, attention should be placed on addressing behavioral health conditions [31]. In Mexico, deported veterans need immediate training in Spanish language and skill development for employment in Mexico. They also need immediate linkage to free and charitable healthcare services, specifically mental healthcare services to cope with loss, grief, and feelings of not belonging related to the trauma of deportation [32].

Third, at the microsystem, deported veterans need to continue their connection with their family and friends in the US and establish new networks in Mexico. Deported veterans experience social stigma not only from being viewed as deported criminals, but also due to their history of US military service given the unfavorable views towards the US in many countries [33, 34]. Like what others have found, veterans in our study perceived themselves as "outsiders in Mexico"—they lacked the confidence, language, and social skills needed to successfully navigate Mexican society [32]. By connecting veterans to community-based peer support groups, especially veteran service organizations, they can find community and identity with other veterans [10]. This was the case for veterans in our research—upon deportation, veterans quickly connected to other deported veterans.

As the narratives of deported veterans in our study illustrate, this precarious situation provoked by deportation penetrates the infrastructure that they navigate reducing their ability to live out daily life and harms their interpersonal relationships, which ultimately create chronic stress and result in poor health.

## Limitations

Several limitations should be considered when interpreting our results. We utilized non-probability sampling methods to recruit participants, which can introduce selection bias. That said, the social determinants of health presented in this study reflect the daily experiences of participants who had been deported to Mexico and were affiliated with the DVSH. Thus, insight provided by our participants may not reflect the experiences of deported US veterans not affiliated with the DVSH, living outside Tijuana, living outside of Mexico, or of a different nationality by birth. It may also not represent the experiences of women deported veterans or those who served in the US Air Force or Coast Guard. These populations of deported veterans may hold different priorities and deportation experiences not captured in this study.

Furthermore, on average, participants in the study had been adjusting to their post-deportation lives for a decade. Thus, our results may not accurately depict the experiences of veterans immediately after deportation. For example, while we inquired about participants' current experiences of housing, food, and employment (in)security, we were unable to capture their experiences immediately after deportation. Yet, from veterans' narratives elicited during semi-structured interviews, we know participants frequently lacked access to food, housing, and employment immediately after deportation. Another limitation of our work is the limited insight into deported veterans' interactions with the US criminal justice system. Future research should explore the intersections between criminality and deportation as it is the criteria for deportation of noncitizen veterans. Our findings reinforce that congressional changes

to criminal justice and immigration policies are needed to restore judicial discretion in veteran deportation cases.

## Conclusion

The US has a total of 19 million veterans [6]. Due to current military, criminal justice, and immigration policies, noncitizen US military veterans, especially those presenting with mental health and substance use disorders which underlie risky and unlawful behaviors, are at heightened risk for deportation. Without change in our nation's policies, noncitizen veterans will continue to present in our immigration system and face deportation charges. Our study provides an important foundation for future work with deported veterans from an academic, journalistic, legal, and policy perspective as they demonstrate the value of eliciting veteran-centered perspectives on how deportation relates to health and wellbeing. During a symposium that displayed deported veterans' photos and narrative text, we engaged policy experts, VA healthcare providers, advocates, and veterans in discussions of next steps in terms of policy change and research. The discussions highlighted the need for continued work to change existing immigrant policies as well as the deportation effects on veterans' health to be studied in future research as a possible social determinant of health.

## Supporting information

**S1 Data. "At the Pawn Shop".**
(DOCX)

**S2 Data. "Hello, my name is Johnny. How may I help you?".**
(DOCX)

**S3 Data. "On the Go in Mexico".**
(DOCX)

**S4 Data. "My Life in Crumbles".**
(DOCX)

**S5 Data. "What Hurts the Most".**
(DOCX)

**S6 Data. "Bleeding Out".**
(DOCX)

**S7 Data. "Taking Away the Pain".**
(DOCX)

**S1 Text.**
(PDF)

## Acknowledgments

The content is solely the responsibility of the authors and does not necessarily represent the official views of the funding sources. We would like to thank Jo Gerrard at the University of California, Riverside School of Medicine for her editorial support.

## Author Contributions

**Conceptualization:** Frances Tao, Cassidy T. Lee, Ann Marie Cheney.

**Project administration:** Edgar Castelan.

**Writing – original draft:** Frances Tao, Cassidy T. Lee, Ann Marie Cheney.

**Writing – review & editing:** Frances Tao, Cassidy T. Lee, Edgar Castelan, Ann Marie Cheney.

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
