## [Decision Letter · Decision Letter 0]

29 Nov 2022

PGPH-D-22-00720

Social Determinants of Health among Noncitizen Deported US Veterans: A Participatory Action Study

Thank you for submitting your manuscript to PLOS Global Public Health. After careful consideration, we feel that it has merit but does not fully meet PLOS Global Public Health’s publication criteria as it currently stands. Therefore, we invite you to submit a revised version of the manuscript that addresses the points raised during the review process.

The topic is of great interest and suitable for publication. However, there is a need of improving the manuscript. Discussion and conclusion should be re-elaborated based on reviewer comments.

The procedure of photovoice should be better explained, as well as the justification of the planned meetings during it. How do the authors intend to help participants to be heard by policymakers? Explain better how the analysis of the images and audio recording were.

I hope you can address these aspects and those indicated by the reviewers to improve the manuscript.

We look forward to receiving your revised manuscript.

Kind regards,

Maria del Mar Pastor Bravo, Ph.D.

Academic Editor

Journal Requirements:

Reviewers' comments:

Reviewer's Responses to Questions

**Comments to the Author**

1. Does this manuscript meet PLOS Global Public Health’s publication criteria? Is the manuscript technically sound, and do the data support the conclusions? The manuscript must describe methodologically and ethically rigorous research with conclusions that are appropriately drawn based on the data presented.

Reviewer #1: Partly

Reviewer #2: No

2. Has the statistical analysis been performed appropriately and rigorously?

Reviewer #1: Yes

Reviewer #2: No

3. Have the authors made all data underlying the findings in their manuscript fully available (please refer to the Data Availability Statement at the start of the manuscript PDF file)?

Reviewer #1: Yes

Reviewer #2: No

4. Is the manuscript presented in an intelligible fashion and written in standard English?

Reviewer #1: Yes

Reviewer #2: Yes

5. Review Comments to the Author

Reviewer #1: Thank you very much for requesting my review on the manuscript tilted " Social Determinants of Health among Noncitizen Deported US Veterans: A Participatory Action Study". This is a topic of major importance from both public health and health equity perspectives.

The methodology was comprehensive in utilizing more than one qualitative tool to get information which gives more validity to the results. Additionally, engaging community in planning is proven to enhance the output with deeper insight due to gaining trust of the target population. However, i have some points to consider in this manuscript to be suitable for publication:

1- Quantitative data utilization. Although, it was declared as a limitation of the study, quantitative data analysis, literature shows that the minimum sample to be considered for quantitative analysis ranges between 20-40 minimum and without a probability sampling. Additionally, the statistical mean comparison between two populations with huge number difference does not provide stable nor robust results even if results would be weighted. Therefore, I would highly recommend to eliminate the quantitative analysis from the manuscript and adjust the results and discussion accordingly. This adjustment should not harm the concept and results since the key findings were extracted from the interviews and photo voice. This study would be highly valued as a qualitative study supported with interviews and photo voice tools.

2- This study with all the effort done by the authors and collaborators, cannot provide a conclusive result that deportation "is a social determinant of health". However, it could suggest deportation as a possible determinant to be studied further in future research. Conclusive results demand different methodologies, sample size and analysis.

Reviewer #2: Background:

The first sentence seems at odds with the rest of the paragraph – perhaps adjust to “The US military may provide a fast track to citizenship….”

I would like to see a better link between the paragraphs in the background. Jumping straight from citizenship to discharge to health consequences doesn’t flow well, I think the authors are endeavouring to link the implications of post-discharge behaviours and potential deportation, and flow on effects resulting from the deportation, it’s just a little clunky as is.

This section would benefit from bridging sentences or paragraph to demonstrate the link between the concepts. Alternatively, an introductory paragraph that announces the concepts and their links before moving into the separate concepts.

Methods:

The “community review board” language is a little confusing in the same paragraph as the IRB – is there a different term that might be used to differentiate their purpose?

Pg 9: The manuscript directs to Table 1 for SF-36 scores and comparisons with the general population. Table 1 does not contain any of the SF-36 survey criteria, nor does it show population comparison. I would also question that the general US population is the best comparison group for a very small military cohort, many of whom experienced life changing events during service that would not occur in the general population, and all of whom were incarcerated.

Table 1 is also confusing in regard to the % column – % is not relevant to some items (eg mean age, age at first entry, year enlisted, year discharged etc) so it is unclear what the number in brackets is in these cases. I think it is referring to SD, but the table needs to be reformatted for clearer understanding.

Results/Discussion:

The methods described to not relate to the results reported.

Some of the language is opinion, subjective, and emotive, and there does not appear to be evidence to back the position taken.

I’m sure there was substantial qualitative data collected, so there may be a better way to report the information/data so that the results are more demonstrable and robust, rather than what appears to be author interpretation.

General: I’m not confident that the data collected on a very discrete group is sufficient to report the findings outlined in this manuscript, especially noting the significant limitations that the authors themselves have identified.

6. PLOS authors have the option to publish the peer review history of their article (what does this mean?). If published, this will include your full peer review and any attached files.

**Do you want your identity to be public for this peer review?** For information about this choice, including consent withdrawal, please see our Privacy Policy.

Reviewer #1: No

Reviewer #2: No

---

## [Decision Letter · Decision Letter 1]

27 Apr 2023

PGPH-D-22-00720R1

Social Determinants of Health among Noncitizen Deported US Veterans: A Participatory Action Study

Dear Dr. Cheney,

Thank you for submitting your manuscript to PLOS Global Public Health. After careful consideration, we feel that it has merit but does not fully meet PLOS Global Public Health’s publication criteria as it currently stands. Therefore, we invite you to submit a revised version of the manuscript that addresses the points raised during the review process.

While the reviewers feel positive about the revisions on the manuscript, before we proceed further, please include a complete copy of PLOS’ questionnaire on inclusivity in global research in your revised manuscript. Our policy for research in this area aims to improve transparency in the reporting of research performed outside of researchers’ own country or community. The policy applies to researchers who have travelled to a different country to conduct research, research with Indigenous populations or their lands, and research on cultural artefacts. The questionnaire can also be requested at the journal’s discretion for any other submissions, even if these conditions are not met. Please find more information on the policy and a link to download a blank copy of the questionnaire here: https://journals.plos.org/plosone/s/best-practices-in-research-reporting. Please upload a completed version of your questionnaire as Supporting Information when you resubmit your manuscript.

We look forward to receiving your revised manuscript.

Kind regards,

Lucinda Shen, MSc

Staff Editor

Journal Requirements:

Additional Editor Comments (if provided):

Reviewers' comments:

Reviewer's Responses to Questions

**Comments to the Author**

1. If the authors have adequately addressed your comments raised in a previous round of review and you feel that this manuscript is now acceptable for publication, you may indicate that here to bypass the “Comments to the Author” section, enter your conflict of interest statement in the “Confidential to Editor” section, and submit your "Accept" recommendation.

Reviewer #2: All comments have been addressed

Reviewer #3: (No Response)

2. Does this manuscript meet PLOS Global Public Health’s publication criteria? Is the manuscript technically sound, and do the data support the conclusions? The manuscript must describe methodologically and ethically rigorous research with conclusions that are appropriately drawn based on the data presented.

Reviewer #2: Yes

Reviewer #3: Yes

3. Has the statistical analysis been performed appropriately and rigorously?

Reviewer #2: Yes

Reviewer #3: N/A

4. Have the authors made all data underlying the findings in their manuscript fully available (please refer to the Data Availability Statement at the start of the manuscript PDF file)?

Reviewer #2: Yes

Reviewer #3: Yes

5. Is the manuscript presented in an intelligible fashion and written in standard English?

Reviewer #2: Yes

Reviewer #3: Yes

6. Review Comments to the Author

Reviewer #2: Thank you for the extensive revisions. You have addressed my concerns with the manuscript.

One pedantic point (that does not effect my recommendation to publish):

- I would still like to see Table 1 a little clearer. eg three columns - Characteristics / n% / Mean (SD)

Reviewer #3: Thank you very much for the opportunity to review this revised manuscript on a participatory action study of noncitizen deported veterans’ experiences and health concerns. I found the revised submission to be highly responsive to reviewer comments on the previously submitted version of the manuscript. My only additional suggestions below are very minor recommended changes/corrections to several wording usages throughout the manuscript – please consider:

1. Throughout the manuscript:

Using only one of “sociodemographic” or “socio-demographic”

2. Throughout the manuscript:

Using only one of “sociocultural” or “socio-cultural”

3. “Semi-structured Interviews” section, second paragraph, first sentence:

Clarifying how the first question probed about health

4. “Semi-structured Interviews” section, second-to-last sentence:

Removing the second use of “interviews” from the sentence

5. Discussion section, first paragraph:

Changing two instances of “see” to “saw”

6. Discussion section, second paragraph, last sentence:

Changing “veterans service history” to “veterans’ service history”

7. Conclusion section, second-to-last sentence:

Inserting a word (“engaged”?) between “we” and “policy” in “we policy experts”

8. Conclusion section, last sentence:

Inserting “be” before “studied”

9. Funding section, before last sentence:

Removing quotation marks

7. PLOS authors have the option to publish the peer review history of their article (what does this mean?). If published, this will include your full peer review and any attached files.

**Do you want your identity to be public for this peer review?** For information about this choice, including consent withdrawal, please see our Privacy Policy.

Reviewer #2: No

Reviewer #3: **Yes: **Bo Kim

---

## [Decision Letter · Decision Letter 2]

26 Jun 2023

Social Determinants of Health among Noncitizen Deported US Veterans: A Participatory Action Study

PGPH-D-22-00720R2

Dear Dr. Cheney,

We are pleased to inform you that your manuscript 'Social Determinants of Health among Noncitizen Deported US Veterans: A Participatory Action Study' has been provisionally accepted for publication in PLOS Global Public Health.

Best regards,

Julia Robinson

Executive Editor

Reviewer Comments (if any, and for reference):

Reviewer's Responses to Questions

**Comments to the Author**

1. If the authors have adequately addressed your comments raised in a previous round of review and you feel that this manuscript is now acceptable for publication, you may indicate that here to bypass the “Comments to the Author” section, enter your conflict of interest statement in the “Confidential to Editor” section, and submit your "Accept" recommendation.

Reviewer #2: All comments have been addressed

Reviewer #3: All comments have been addressed

2. Does this manuscript meet PLOS Global Public Health’s publication criteria? Is the manuscript technically sound, and do the data support the conclusions? The manuscript must describe methodologically and ethically rigorous research with conclusions that are appropriately drawn based on the data presented.

Reviewer #2: Yes

Reviewer #3: Yes

3. Has the statistical analysis been performed appropriately and rigorously?

Reviewer #2: Yes

Reviewer #3: N/A

4. Have the authors made all data underlying the findings in their manuscript fully available (please refer to the Data Availability Statement at the start of the manuscript PDF file)?

Reviewer #2: Yes

Reviewer #3: Yes

5. Is the manuscript presented in an intelligible fashion and written in standard English?

Reviewer #2: Yes

Reviewer #3: Yes

6. Review Comments to the Author

Reviewer #2: While I am happy to accept this article as is, you might consider changing all instances of "Latino" to 'Latinx' prior to publication to ensure gender neutral language.

Pg, ln 1, please change i'indicate' to "indicated'.

Thank you for addressing my other comments.

Reviewer #3: I sincerely thank the authors for very thoroughly incorporating the previously suggested edits, and I have no further comments to offer to the authors.

7. PLOS authors have the option to publish the peer review history of their article (what does this mean?). If published, this will include your full peer review and any attached files.

**Do you want your identity to be public for this peer review?** For information about this choice, including consent withdrawal, please see our Privacy Policy.

Reviewer #2: **Yes: **Rachelle Warner

Reviewer #3: **Yes: **Bo Kim
